# Whole-genome sequencing analysis of anthropometric traits in 672,976 individuals reveals convergence between rare and common genetic associations

Gareth Hawkes [1,4] ✉, Harrison I. W. Wright [1,4], Robin N. Beaumont [1,4], Kartik Chundru [1,4], Aimee Hanson [2], Leigh Jackson [1], Anna Murray [1], Kashyap Patel [1], Timothy M. Frayling [3], Caroline F. Wright [1,4], Andrew R. Wood [1,4] ✉ & Michael N. Weedon [1,4] ✉

GWAS have generally focused on common variants from genotyping arrays or rare protein-coding variants from exome sequencing. Here, we use whole-genome sequencing data to evaluate the contribution to and architecture of rare non-coding variants for three commonly studied anthropometric traits: height, BMI and waist-hip ratio adjusted for BMI. Analysing 447,461 individuals in the UK Biobank for discovery and 225,515 individuals in All of Us for replication, we identify 90 rare and low-frequency single variant associations, including two independent rare variants upstream of *IGF2BP2* that substantially reduce waist-hip ratio adjusted for BMI, but have distinct effects on other adiposity traits. We further identify 135 coding variant aggregates. For example, *UBR3* protein-truncating variants are associated with a 2.7 kg/m2 increase in BMI. We additionally identify 51 non-coding variant aggregate associations, including one in the 5'UTR of *FGF18* associated with up to 6 cm effects on height. We show that 97% of rare variant associations occur near GWAS-identified loci, demonstrating convergence of rare and common variant associations. Finally, we show that ultra rare variants explain a small fraction of heritability compared to common variants for these traits, that heritability is largely shared across ancestries, and that it concentrates around common variant loci.

Human genetic association studies have mostly focused on common variants from genome-wide association studies (GWAS), or rare protein-coding variants from exome sequencing studies. Despite the successes of these studies, a substantial amount of heritability remains unaccounted for[1]. For example, a GWAS of 5.4 million individuals identified 12,111 common variant associations for adult height[2], which accounted for ~40% of the variance among individuals of European genetic ancestry. However, the total heritability for height is estimated

[1]Department of Clinical and Biomedical Sciences, Faculty of Health and Life Sciences, University of Exeter, Exeter, United Kingdom. [2]Medical Research Council Integrative Epidemiology Unit, Department of Population Health Sciences, Bristol Medical School, University of Bristol, Bristol, United Kingdom. [3]Faculty of Medicine, Department of Genetic Medicine and Development, CMU, University of Geneva, 1 rue Michel-Servet, Geneva, Switzerland. [4]These authors contributed equally: Gareth Hawkes, Harrison I. W. Wright, Robin N. Beaumont, Kartik Chundru, Caroline F. Wright, Andrew R. Wood, Michael N. Weedon. ✉e-mail: g.hawkes2@exeter.ac.uk; a.r.wood@exeter.ac.uk; m.n.weedon@exeter.ac.uk

to be ~80%[3], indicating that many additional genetic factors remain to be discovered.

Rare non-coding variants may explain a substantial amount of the remaining heritability, but their contribution is largely unexplored. Only a small number of prior studies have estimated rare-variant heritability using WGS data, typically limited to small sample sizes due to data constraints[4,5]. The causal variants and effector genes at most GWAS loci have not been identified using imputed and exome sequencing data because of small effect sizes, extensive linkage disequilibrium (LD) and because most associations are non-coding. Larger effect sizes and limited LD mean that rare variants have substantial advantages over common variants in identifying causal variants and genes. Rare non-coding associations will therefore allow us to identify key regulatory elements for genes important in human biology and disease[6]. Identifying rare non-coding associations may also substantially aid the identification of the causal gene at GWAS loci.

There are many examples of genes where rare coding variants cause monogenic disease, while common, usually non-coding, variants subtly predispose to a related trait. For example, loss-of-function variants in *MC4R* cause severe obesity[7], whereas distal non-coding regulatory variants are associated with a small increase in BMI[8]. A spectrum of associations from common low-effect to rare large-effect variants at a locus (an 'allelic series') can substantially aid in the identification of the effector gene at a locus[9]. It also provides the opportunity to assess the effect of differential levels of disruption on a gene, which can provide key insights into biological mechanisms and can be important in drug development[10]. How often rare and common variant associations converge on the same genes[11] and regulatory elements is therefore a key unanswered question.

WGS-association analyses in 100,000's of individuals are now becoming possible. For example, the UK Biobank (UKB) has released WGS data from 500,000 individuals[12] and, at the date of writing, the All of Us study[13,14] (AoU) has WGS data available on nearly 250,000 individuals. We previously used an interim release of WGS data on 200,003 individuals from UKB to develop a framework for WGS-based association testing[15]. We discovered and replicated rare variant associations missed by single variant-based GWAS and exome sequencing studies. For example, we identified rare non-coding aggregate-based associations for height proximal to *HMGA1* and *miRNA-497*.

Here, we substantially extend our previous study by analysing three key anthropometric traits that have been the central focus of previous large-scale GWAS meta-analyses: adult height, body-mass index (BMI), and waist-hip-ratio adjusted for BMI (WHRadjBMI) using annotated variants from WGS data on up to 447,461 (mean age = 57.3 years, $N$-female = 242,966) individuals of inferred European (EUR) genetic ancestry from the UKB, a population cohort from the United Kingdom. As our interest was in rare variants, the two second-largest inferred ancestries within UKB were not suitable for our primary discovery analyses due to their small sample size: UKB South-Asian (UKB-SAS; $N$ = 9901, mean age = 54.0 years, $N$-female = 5185) and UKB African (UKB-AFR; $N$ = 7788, mean age = 52.4 years, $N$-female = 4513). For our primary analysis, we decided against a cross-ancestry approach in the UK Biobank due to concerns regarding population stratification effects, particularly for rare variants[16]. Instead, to demonstrate the generalisability of our findings beyond EUR-individuals, we replicated our results in the diverse AoU study, based on a meta-analysis of 225,515 individuals of diverse genetic ancestry from the AoU study of inferred European (AoU-EUR; $N$ = 128,566, mean age = 56.0 years, $N$-female = 77,907), African (AoU-AFR; $N$ = 54,940, mean age = 49.7 years, $N$-female = 32,117) and admixed-American (AoU-AMR; $N$ = 42,009, mean age = 45.5 years, $N$-female = 28,317) genetic ancestry. However, a meta-analysis of genetically inferred UKB-EUR (our primary discovery cohort), UKB-SAS and UKB-AFR was performed as a secondary analysis for comparison.

We performed both single variant (minor-allele count (MAC) ≥ 5) and genomic aggregate association tests (minor-allele frequency (MAF) < 0.1%) using REGENIE[17]. Phenotypes were adjusted for age, age squared, sex, recruitment centre, WGS centre and 40 genetic principal components at runtime (**"Methods"**). We annotated all genetic variants using Ensembl's Variant Effect Predictor (VEP)[18] v110 (**"Methods"**) and used the output to categorise variants as gene-centric (e.g., coding, predicted intronic splicing, intronic unspecified, proximal-regulatory) and intergenic-regulatory (e.g., Ensembl regulatory regions, non-coding RNA, intergenic unspecified) for aggregate-based association testing. Additionally, we performed aggregate testing on all non-coding variants in overlapping (1 kbp overlap) 2 kbp sliding windows. We also sub-categorised variants within a subset of aggregate units by measures of constraint (JARVIS[19]), conservation (GERP[20]) and/or predicted deleteriousness (CADD[21]). We applied three rare-variant aggregate testing procedures built into REGENIE: BURDEN, where all variants are assumed to act homogeneously on the phenotype; SKAT, where variants may act bi-directionally; and ACAT, where variants can act both bi-directionally, and some variants can have null estimates (see **"Methods"** for more details).

To identify conditionally independent genetic variants in our primary discovery analysis using UKB-EUR, we used a modified version of GCTA-COJO[22] with the UKB-EUR sequencing data as the reference panel. To identify statistically independent rare-variant aggregates, we performed a forward-stepwise conditional approach (**"Methods"**). Using these methodologies, we were unable to identify conditionally independent associations for the secondary UKB-meta-analysis due to concerns regarding cross-ancestry pooling for LD-panels[2], particularly for UKB sequencing data applied to rare variant association discovery.

We identify hundreds of rare non-coding and rare protein-coding variant associations; identify many rare non-coding variant associations that can only be identified from WGS analysis; demonstrate the power of WGS analyses to fine-map common variant associations; and show that, for these traits, most rare variant heritability co-localises with GWAS loci.

## RESULTS

### 160 rare and low-frequency variants associated with anthropometric traits in UKB-EUR after adjusting for previously associated variants

We identified a total of 2690 statistically independent single-variant associations in our UKB-EUR-based analysis for height, BMI and WHRadjBMI reaching $P < 3 \times 10^{-10}$ (Supplementary Fig. 1). Of the 2690 genetic associations, 91 were rare (MAF < 0.1%) and 114 were low frequency (0.1% ≤ MAF < 1%) (Supplementary Data 1). After adjustment for variants previously reported to be associated with the respective trait by the GIANT consortium[2,23–27], 215 (8%) single-variant associations remained significant, of which 160 (74.4%; 137 for height, 13 for BMI and 10 for WHRadjBMI) were low-frequency or rare (MAF < 1%; Supplementary Data 2). In our secondary all-ancestry UKB meta-analysis, we observed a high degree of loci overlap: 95% of study-wide-significant meta-variants were within 1MB of a variant identified in the primary UKB-EUR analysis (**Data Availability**).

### Single-variant associations replicate in AoU with effect sizes consistent across ancestries

We attempted to replicate our results using 225,515 individuals from the AoU study (N EUR = 128,566; N AFR = 54,940; N AMR = 42,009). Of the 215 (common, low-frequency and rare) single-variant associations in UKB, 172 were available for analysis with minor-allele counts (MAC) ≥ 5 in one or more of the three broad genetic-ancestry groups in AoU (42 with UKB-MAF < 0.1%; 77 with 0.1% ≤ UKB-MAF < 1%; 53 with UKB-MAF ≥ 1%; Supplementary Data 3). Of the 119 rare and low-frequency variant associations that we could put forward for replication, 90 (75.6%) showed nominal evidence of replication ($P < 0.05$), and 42 (35.3%) showed strong evidence of replication at $P < 0.05/172$ with consistent effect sizes. Of the 53 common variants with UKB-MAF > 1%

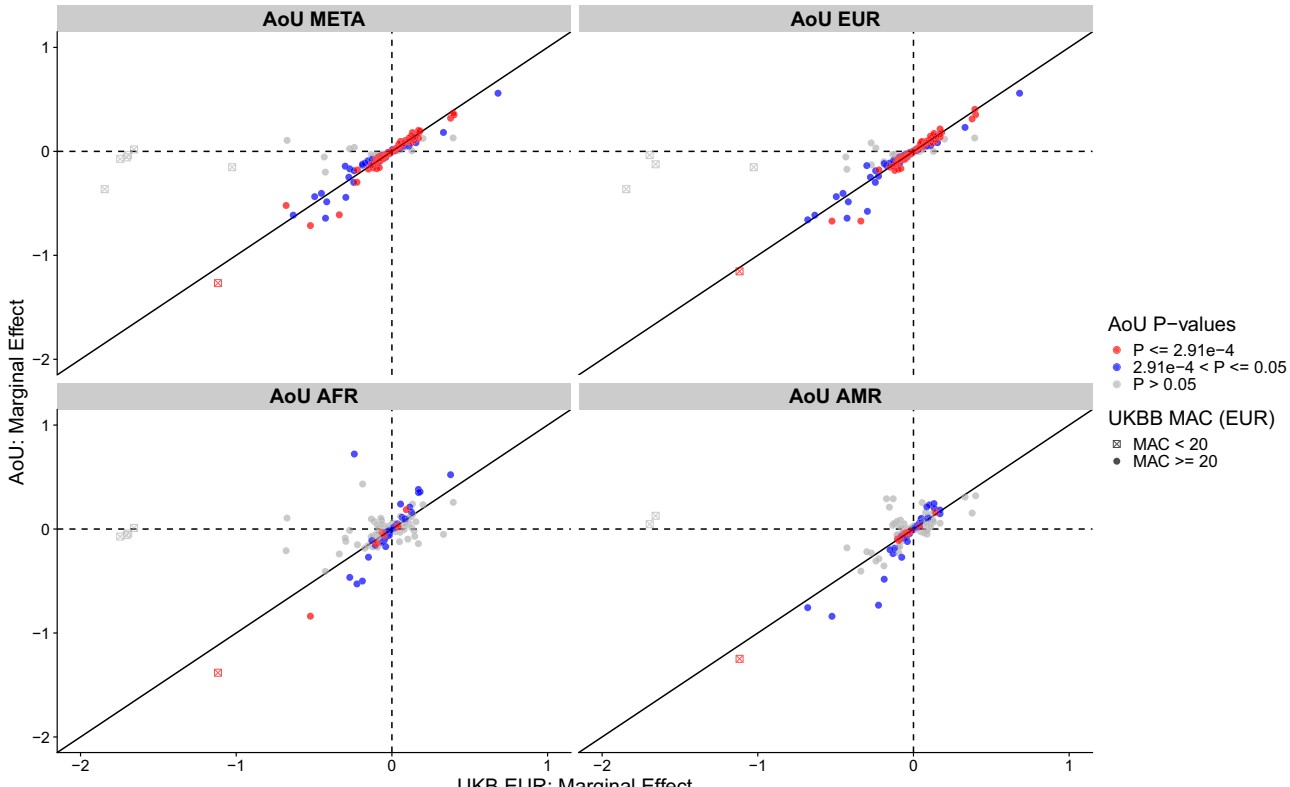

**Fig. 1 | Replication of UK Biobank single variant associations in All of Us.** Comparison of single variant effect estimates from discovery (UK Biobank European; UKB-EUR) to replication analyses (All of Us European, Admixed-American and African; AOU-EUR, AOU-AMR and AOU-AFR respectively). Points are coloured according to their replication p-value (Bonferroni $P \leq$ 2.91e-4 or nominally significant 2.91e-4 $< P \leq$ 0.05, versus null $P >$ 0.05) based on a two-sided chi-squared test, given we performed 172 tests. Circles represent cases where UKB-EUR minor-allele count (MAC) $>$ 20, and squares represent variants where UKB-MAC $\geq$ 5. Only variants with MAC $\geq$ 5 in at least one AoU ancestry were put forward for replication.

put forward for replication, we observed 47 (88.7%) associations with nominal evidence of replication ($P <$ 0.05), and 31 (58.5%) with Bonferroni-corrected evidence of replication ($P <$ 0.05/172). No variants with UKB-MAC $\leq$ 10 (7/172; 4.07%) showed even nominal evidence of replication. However, we observed strong directional consistency in effect sizes for all single variants reaching $P <$ 0.05 across ancestries (Fig. 1: N directionally consistent = 137/138; 99.3%; Binomial $P = 7.53 \times 10^{-177}$). Fifteen percent (33/215) of these variants would not have been detected by conventional GWAS and imputation-based approaches because they were not present in the Haplotype Reference Consortium + UK10K, TOPMed, or Genomics England imputed UKB datasets. Of the remaining 182 (85%) variants, the imputation quality across the imputation panels was variable (Supplementary Data 4).

### A single rare variant in the seed region of *MIRNA497* affects height by 4 cm

We identified a rare insertion (GRCh38:17:7017939:C:CT; MAF = 0.014%) in *MIRNA497* ($P = 6.53 \times 10^{-16}$), with a per-allele effect of 3.79 cm [95% CI 2.87, 4.71 cm] on height in UKB. This association was replicated in AoU ($P = 1.38 \times 10^{-27}$). This variant partially explains an aggregate-based association we previously discovered with highly conserved variants in and near miRNA host-gene *MIRNA497HG* and its by-products *MIR195* and *MIR497*[15]. However, this insertion is statistically independent of an aggregate-based association of highly-conserved (GERP > 2) variants overlapping the promoter region of host-gene *MIR497HG* ($P = 7.17 \times 10^{-12}$), i.e., the remainder of the signal. Our results suggest that regulation of the miRNA497 product impacts height independently of the coding consequences, acting via the rare insertion.

### Two independent rare variants upstream of *IGF2BP2* associate with WHRadjBMI

We identified two independently-associated rare single variants upstream of *IGF2BP2* (which was the closest gene) associated with WHRadjBMI: GRCh38:3:185826396:AG:A (beta = −0.17 SD [−0.22, −0.12], MAF = $1.59 \times 10^{-3}$, $P = 2.12 \times 10^{-11}$, replication $P = 2.24 \times 10^{-2}$) and GRCh38:3:185847637:G:A (beta = −0.12 SD [−0.16, −0.09], MAF = $3.10 \times 10^{-3}$, $P = 2.99 \times 10^{-11}$, replication $P = 4.19 \times 10^{-4}$). Although both variants associate strongly with WHRadjBMI, they have heterogeneous effects on other phenotypes. For example, the former occurs in an lncRNA (TCONS_00006340) previously linked to fasting glucose[28] and is associated with favourable adiposity and height, whilst the latter is not (Fig. 2; Supplementary Data 5), suggesting different regulatory effects on *IGF2BP2*.

### WGS identifies protein-coding aggregate-variant associations missed by exome sequencing

We identified 135 independent rare coding variant (MAF < 0.1%) aggregate associations (Supplementary Data 6), of which 125 associations (height: 114; BMI: 8; WHRadjBMI: 3) remained significant after adjustment for variants previously reported by the GIANT consortium as associated with the respective trait (Supplementary Data 7). The set of genic aggregate-based associations included 73 predicted loss-of-function (58.4%), 49 missense (39.2%), two splice region (1.60%) and one synonymous (0.80%). We put forward 118 of the coding aggregate-based associations for replication in AoU, where there were sufficient numbers of carriers, defined as at least five minor-allele carriers in any of the three largest ancestral groups. Of the 118 associations, 91 showed nominal ($P <$ 0.05) evidence of replication, and 45 passed a

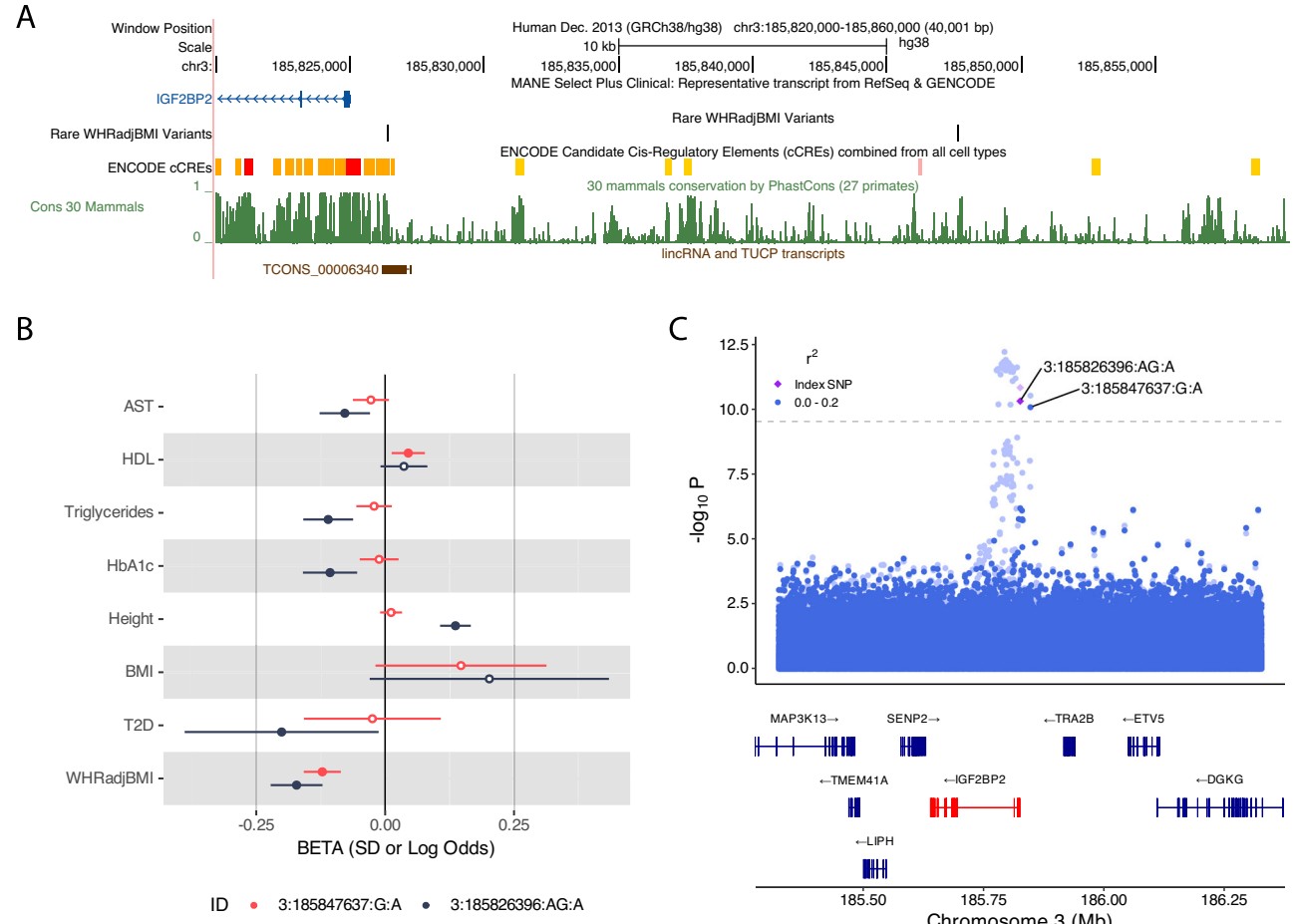

**Fig. 2 | Summary of genetic associations at the *IGF2BP2* locus with WHRadjBMI.**
**A** UCSC genome browser plot showing the location of the two rare variants relative to *IGF2BP2*, a measure of conservation (Cons 30 Mammals) and the overlapping lincRNA. **B** Forest plot showing the effect size of 3:185847637:G:A and 3:185826396:AG:A against metabolic-related phenotypes (*N* = 447,461). Error bars represent the 95% confidence-intervals, centred on the effect size estimate.
**C** LocusZoom plot showing all variants unadjusted (pale background) and adjusted for common (>1% MAF) lead variants (solid foreground), centred on the lincRNA-overlapping variant (purple diamond = 3:185826396:AG:A).

Bonferroni-corrected *P*-value threshold of $P < 4.24 \times 10^{-4}$ (Supplementary Data 8). The one synonymous aggregate association (in *IRS4*, associated with height) did not replicate, suggesting a likely false-positive result.

Of the 91 aggregate-based associations showing nominal evidence of association in AoU, we noted that some protein-coding aggregate-based associations had not been identified in previous exome-based association studies. These included predicted loss-of-function variants in *LCORL* ($P = 2.98 \times 10^{-60}$, *P* replication = $8.38 \times 10^{-20}$), *PCSK5* ($P = 1.19 \times 10^{-19}$, *P* replication = $3.76 \times 10^{-3}$), *AMOTL2* ($P = 2.21 \times 10^{-9}$, *P* replication = 0.02), *BCL9* ($P = 5.69 \times 10^{-9}$, *P* replication = 0.02), *MAZ* ($P = 1.24 \times 10^{-10}$, *P* replication = 0.01), and *TRPC4AP* ($P = 3.41 \times 10^{-12}$, *P* replication = 0.04) associated with height. We also observed coding aggregate-based associations for predicted highly-deleterious missense variants (CADD > 25) in *AXL* ($P = 5.39 \times 10^{-11}$, *P* replication = $5.72 \times 10^{-4}$), *LRRC58* ($P = 1.21 \times 10^{-15}$, *P* replication = 0.01), and *SMAD3* ($P = 3.87 \times 10^{-9}$, *P* replication = 0.01).

We further identified an association between predicted loss-of-function variants in *UBR3* and BMI ($P = 1.21 \times 10^{-9}$, *P* replication = $3.12 \times 10^{-3}$) that was not reported by recent UKB exome sequencing association studies. Although the *UBR3*-BMI association was strongest in a *SKAT* framework, which accounts for bi-directional effects, we also observed a genome-wide significant burden (additive) effect (beta = 2.64 kg/m² [1.78, 3.51], $P = 2.20 \times 10^{-9}$).

## We identified 51 non-coding aggregate associations across anthropometric traits, which were particularly enriched in 5′UTRs

We identified 53 independent rare-variant aggregate non-coding associations (Supplementary Data 9), 51 of which remained significant after adjusting for loci previously reported by the GIANT consortium (Supplementary Data 10): 26 for height and 25 for BMI. In total, we identified six 5′ untranslated regions (5′UTR; 11.8%) and four upstream (7.84%) associations, as compared to two 3′UTR (3.92%) and three downstream (5.88%). We additionally identified three ENSEMBL regulatory-region aggregate associations and four regions annotated as RNA. We observed an enrichment of 5′ UTR associations compared to the frequency of all 5′UTR aggregates tested (background = 2.78%; Fisher's Exact OR = 4.66, $P = 2.85 \times 10^{-3}$). Of the 51 associations, we were able to test 50 for evidence of replication in All of Us (Supplementary Data 11): six showed nominal evidence of replication, and three replicated at a Bonferroni-corrected threshold ($P < 0.05/50$). In our secondary all-ancestry UKB meta-analysis, we observed a high degree of loci overlap: 94% of coding genes identified in the UKB meta-analysis were also observed in the primary UKB-EUR analysis (**ST12**). We did not compare statistically significant non-coding aggregates identified in the UKB−meta-analysis (**ST13**) due to our inability to condition on nearby coding variation for meta-analyses.

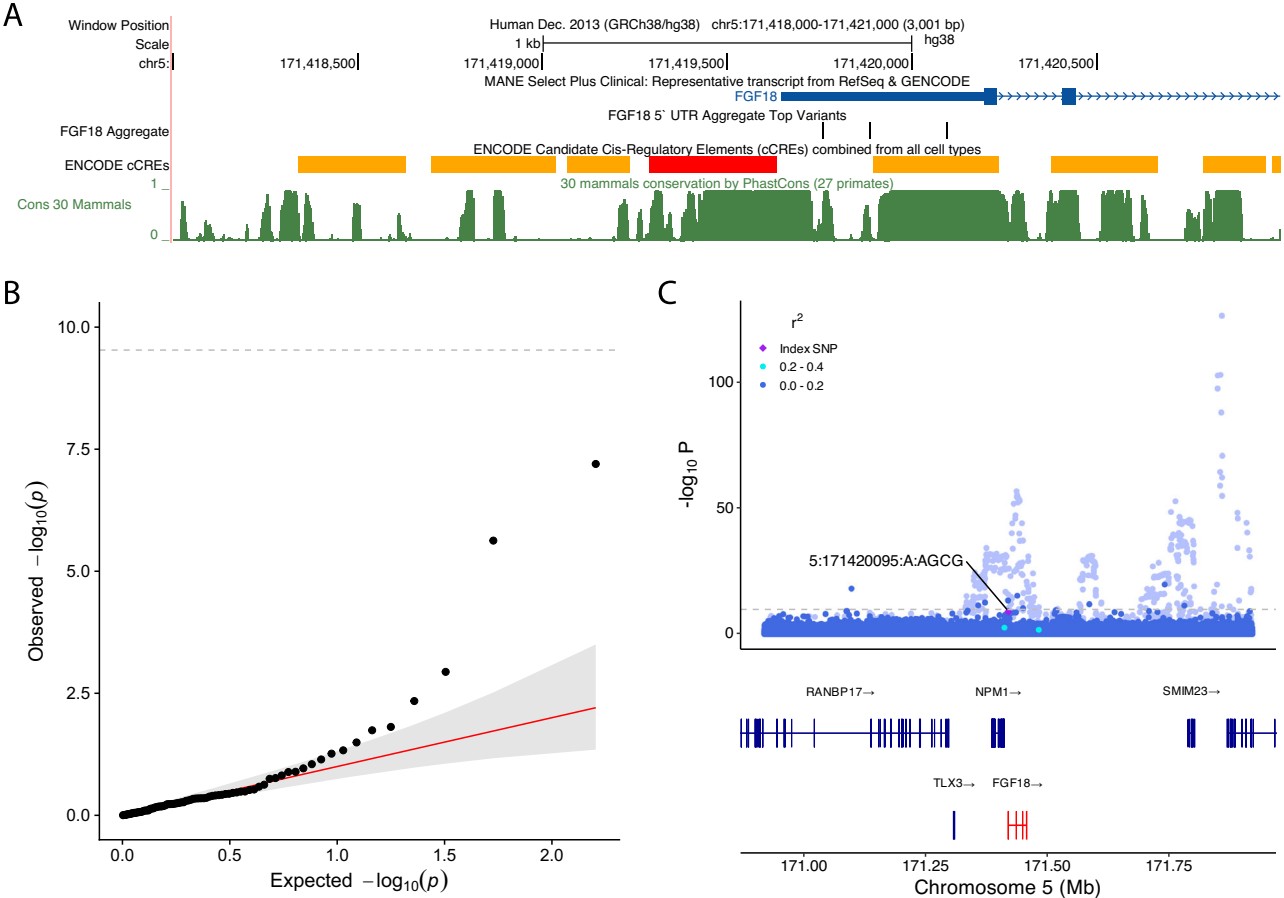

**Fig. 3 | Summary of association between rare variants in the 5'UTR of *FGF18* and height. A** UCSC genome browser plot showing the *FGF18* region, highlighting the position of the most strongly associated single variants in the aggregate, ENCODE cCREs and Conservation (Cons 30 Mammals). **B** QQ plot of variants contributing to the *FGF18* proximal association. The red line shows the expected *p*-value, and the grey band shows the 95% confidence interval. **C** LocusZoom plot showing all variants unadjusted (pale background) and adjusted for common (>1% MAF) lead variants (solid foreground), centred on the most strongly associated variant in the aggregate (5:171420095:A:AGCG).

## Rare 5'UTR variants in *FGF18* substantially influence height

We identified a non-coding aggregate-based association between 5'UTR variants in *FGF18* and height in UKB ($P = 2.93 \times 10^{-12}$), with evidence of replication in AoU ($P = 1.16 \times 10^{-4}$). In our analysis, coding variants of any consequence in *FGF18* were not associated with height ($P \geq 2.12 \times 10^{-3}$), and associations were also absent in both Genebass[29] and the Astrazeneca PheWAS portal[30]. Further analyses suggested that this aggregate-based association was not driven by any single variant (single variant association $P \geq 6.34 \times 10^{-8}$). Within the aggregate, we observed eight variants with MAC ≥ 5, presenting with effect sizes ranging from 6.71 cm to −2.94 cm ($P < 0.05$; Supplementary Data 12; Fig. 3). Four of those eight variants were located at a single multi-allelic position (GRCh38:5:171420095), overlapping an enhancer. Previous GWAS efforts have highlighted seven statistically independent common (MAF > 1%) variants in a cross-ancestry meta-analysis[2] with *FGF18* as their closest gene, and effect sizes ranging from −0.26 cm to 0.13 cm. Together, these results provide a clear example of using rare-variant aggregate testing for the purposes of fine-mapping previously reported GWAS signals.

## Rare single variant and aggregate-variant associations with anthropometric traits usually occur near common variant GWAS loci

We assessed the proximity of rare single variant and aggregate-variant height associations with common variants identified from GWAS. We focused on height primarily because common variant heritability has

been saturated for European individuals as presented in Yengo et al. 2022[2], and secondly, due to the large number of rare single variant and aggregate-variant associations. We assessed proximity to 12,111 common variant associations that were reported to saturate common SNP-based heritability in Europeans and span ~21% of the genome based on 70 kb windows centred on sentinel SNPs (± 35 kb)[2]. Of the 75 variant associations with UKB-EUR MAF < 1% that remained statistically significant in our European-based UK Biobank analysis and with $P < 0.05$ in AoU-EUR, 64 (85%) and 73 (97%) resided within 35 kb and 100 kb (~44% of the genome), respectively (Fig. 4). These observations were similar for 35 common variants that remained statistically significant in our UKB-EUR-based analysis with 32 (91%) and 33 (94%) residing within 35 kb and 100 kb of the 12,111 SNPs, respectively (Supplementary Data 13). Of the 73-coding aggregate-based associations that remained statistically significant in our UKB-EUR-based analysis and with $P < 0.05$ in AoU-EUR, 61 (84%) and 70 (96%) resided within 35 kb and 100 kb of the 12,111 SNPs, respectively. Of the seven autosomal non-coding aggregate-based associations that remained and with $P < 0.05$ in AoU-EUR, all were located within 35 kb of a previously reported common variant (Supplementary Data 14). For BMI, we observed similar levels of physical proximity for four autosomal variants reaching genome-wide significance after adjustment for previously published BMI associations by the GIANT consortium and reaching $P < 0.05$ in AoU-EUR. All four variants (including two with UKB-EUR MAF < 1%) were within 35 kb of 941 common variants previously associated with BMI[26] (Supplementary Data 15). However, two aggregate-based associations

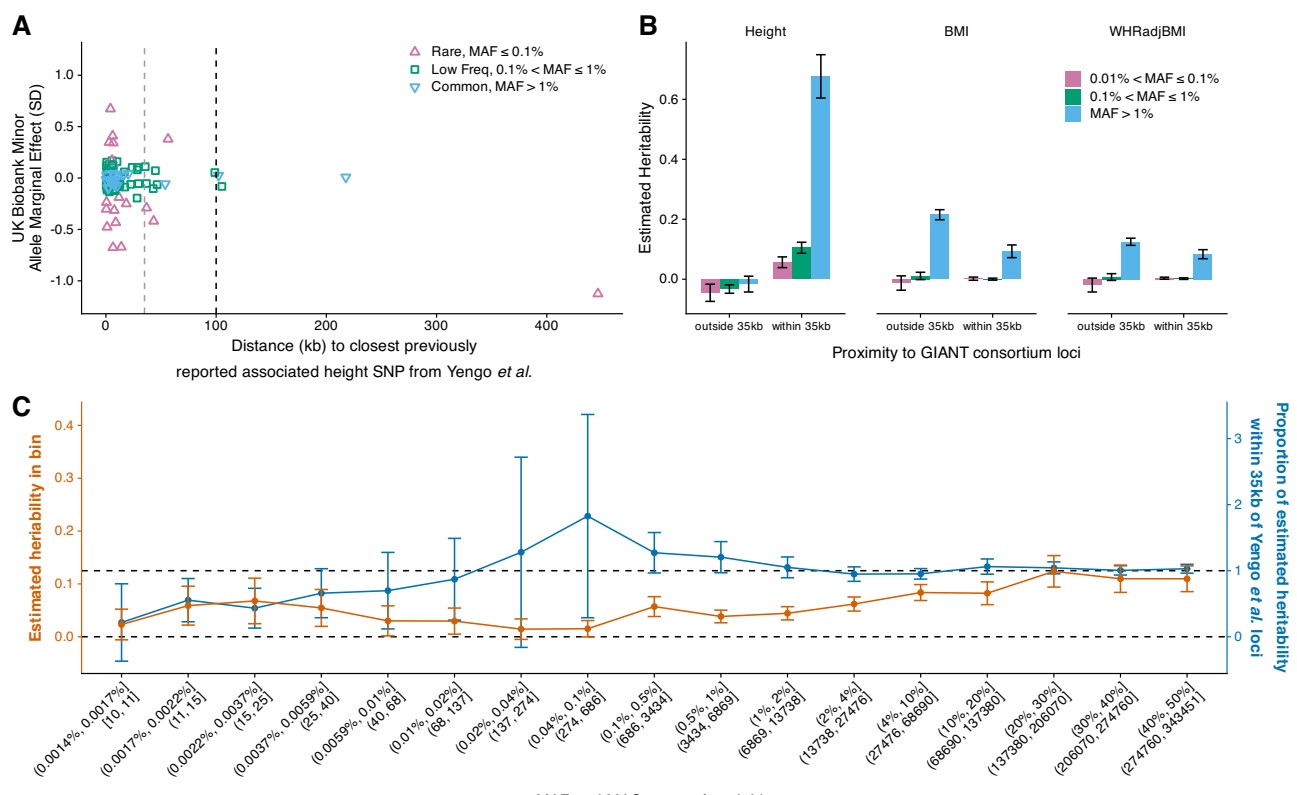

**Fig. 4 | Co-localisation and heritability analyses of anthropometric traits. A** Co-localisation of rare (MAF < 1%) and common (MAF > 1%) independent single variants associated with height in our analysis, relative to those presented in the common-variant heritability saturation *Yengo* et al. *2022*[2]. **B** Estimates of heritability for three variant-frequency bins within and outside of 35 kbp of common variants reported by the most recent GIANT analysis of height, BMI and WHRadjBMI (*N* = 343,451).

**C** Semi-continuous estimation of heritability for height across 17 variant frequency bins (N = 343,451), in contrast to the proportion of heritability for each bin within 35 kbp of a *Yengo* et al. *2022*[2] variant. Error bars represent 95% confidence-intervals throughout, estimated using 100 jackknife blocks and centred on the estimated heritability.

identified in UKB and with AoU-EUR *P* < 0.05 were located further away from the nearest common BMI GWAS SNP (>1.7 Mb) (Supplementary Data 16).

### Rare single variant and aggregate-variant anthropometric trait associations converge across ancestries

Next, we assessed the proximity of genetic associations for height in other broad genetic ancestries, using 54,940 and 42,009 individuals of African (primarily African-American) (AFR) and admixed-American (AMR) genetic ancestry, respectively, in AoU. In our AFR-based analysis, we identified 52 autosomal variants associated with height, of which 45 (87%) reside within 35 kb of a previously reported common variant, including 3/5 associated variants with AoU-AFR MAF < 1%. We also observed 49/52 (94%) associated variants reside within 100 kb, including 4/5 (80%) with AoU-AFR MAF < 1% (Supplementary Data 17). A similar pattern of physical colocalization was observed in our AMR-based analysis. Of the 35 autosomal variants identified as associated with height, 30 variants (86%) reside within 35 kb of a previously reported common variant, including 2 of 3 (67%) with AoU-AMR MAF < 1%. In addition, 34/35 (97%) reside within 100 kb of a previously reported common variant, including all associated variants with AoU-AMR MAF < 1% (Supplementary Data 17).

### Heritability explained by rare variants is enriched around common signals of association

To determine whether rare and common genetic signals of association are likely to converge independently of the genetic associations we had power to detect, we performed partitioned heritability analyses using RHEmc (**"Methods"**). Using a maximally unrelated subset

(common-variant GRM off-diagonal < 0.05) of the UKB-EUR WGS-based analysis (*N* = 343,451, mean age = 57.2 years, *N*-female = 186,241), we observed a total h2 estimate of 0.744, 0.309, and 0.202 (se = 0.036, 0.013, 0.016), of which 11.2%, 0.1%, and −3.4% (h2 = 0.084, 0.0004, −0.007 (se = 0.021, 0.014, 0.013)) is explained by low-frequency and rare variants (UKB-EUR MAF < 1%), for height, BMI, and WHRadjBMI, respectively.

We compared a down-sampled iteration (N = 24,932, mean age = 57.4, *N*-female = 13,332) of our RHEmc results for height with those of Wainschtein et al. (2022)[4], who estimated the heritability for height using 25,465 samples with WGS data in TOPMed. We additionally replicated their methodology by applying GREML implemented in GCTA[22] (**"Methods"**) to the down-sampled data, finding consistent results (Supplementary Fig. 2; Supplementary Data 18).

Of the heritability explained by the low-frequency and rare variants, 100% can be attributed to the variants residing within 35 kb of a previously reported common variant associated with height[2] (Fig. 4, Supplementary Data 19). This value was lower for BMI and WHR adjusted for BMI (31% and 44% respectively), which may reflect the potential exclusion of 35 kb windows around common GWAS signals that are yet to be detected through larger-scale GWAS meta-analyses for these traits.

Finally, we performed a secondary heritability analysis for height to additionally stratify by allele frequencies of variants in the WGS data. We used 17 minor-allele frequency bins, down to and including variants with a MAC of 10, and three LD bins per frequency bin (Fig. 4, Supplementary Data 20; **"Methods"**). The estimate for the proportion of heritability within 35 kb of a previously reported height GWAS SNP was > 0.8 for 12/17 bins, and > 0.99 for 9/17 bins. However, the total

heritability estimate summed across all 17 frequency bins was inflated (h2 = 1.005, SE = 0.0535), with inflation observed across rare-frequency bins (potentially indicative of residual population stratification). To quantify the inflation in the ultra-rare (MAF < 0.0001) variant bins, we fitted an exponential decay model on the cumulative heritability of the bins used in the primary analysis (MAF > 1e-4; Supplementary Fig. 3, Supplementary Data 21). Our model gave an estimate of h2 for all variants with MAC ≥ 10 of 0.79, which implies an inflation of 0.22 from the ultra-rare bins. We estimate a cumulative h2 of 0.76 at MAF 0.0001 and 0.81 at MAC 1, suggesting an ultra-rare variant heritability of approximately 0.05.

Together, these results indicate that rare variant associations are most likely to converge with common genetic signals of association for anthropometric traits.

## Discussion

Here, we use large-scale WGS association analyses to discover hundreds of new rare variant associations for anthropometric traits. We identify rare single variants that have been missed by previous imputation-based GWAS studies. Using rare variant aggregates, we then identify gene-phenotype associations missed by exome sequencing studies and provide clear examples of non-coding aggregate-variant associations for genes that have no coding associations. Finally, our results suggest that rare single variant, coding and non-coding variant aggregates converge on the same genes.

We demonstrate that the majority of previously unreported rare variants associated with height in an analysis of WGS data co-localise with common variants reported by the most recent meta-analysis of imputed data, which reported saturation of the common-variant heritability among individuals of European ancestry. We further show that this result extends to other genetic ancestries, which did not experience a similar heritability saturation. In previous work, Wu et al.[31] inferred through simulations that co-localisation patterns of 35 kb to causal variant for >80% of signals only applied when simulating within frequency bins, i.e., they observed common variants to reside close to common causal variants and rare signals to reside close to rare causal variants. Our work demonstrates that both common and rare variant associations converge on the same genomic regions and genes. This finding is important because it demonstrates that the underlying biological pathways are likely to be similar, and we are therefore assessing the allelic series of effects on genes. Our results also suggest these pathways are consistent across genetic ancestries. Finally, restricting the hypothesis-space for an analysis to the region around known GWAS loci could substantially reduce both multiple-testing thresholds and the environmental impact of computing resources required to analyse these large-scale genetic data[32].

Our work provides another clear example of a non-coding association near a gene that lacks coding associations, specifically, the association of rare 5'UTR variants in *FGF18* with height. This is a small, constrained gene (pLI = 0.98; missense Z = 2.44)[33] with only three protein-truncating variants in the UK Biobank, none of which occur in the last exon, where they would be expected to escape nonsense-mediated decay. This result is similar to a previous finding in *HMGA1*, where we previously showed that rare regulatory variants affect height by up to 5 cm, but there is no protein-coding association for the gene. Coding associations in both these genes may be undetectable, either due to lack of power or because haploinsufficiency (or other coding effects) is incompatible with life, whilst more subtle effects on expression are compatible with life and result in a detectable effect on height. Both *FGF18* and *HMGA1* have common variant GWAS signals, and our work demonstrates that these are the likely causal genes.

We further demonstrate that aggregate testing is a powerful way to identify associations with rare coding and non-coding variants. We previously identified *MIRNA497* as being associated with height using an aggregate-based association in 200,000 WGS in the UK Biobank[15].

With increased statistical power we now show a single rare deletion in the seed region of *MIRNA497* is independently associated with height and is a likely causal variant at the locus. Our results now suggest that some variants influence the expression of the *MIRNA497* and others independently influence height by affecting its ability to bind targets. There is substantial orthogonal evidence for an important role of *MIRNA497* in human growth[34–36].

We identify several anthropometric trait gene associations missed by existing exome sequencing studies. We found a substantial number of coding aggregate associations for height. Most of these missed associations were because of low coverage in the exome sequencing compared to genome sequencing. For example, one of the most strongly associated genes is *LCORL*, a GWAS locus for height. In GeneBass and Astrazeneca PheWAS portal, *LCORL* PTVs are not associated with height (P > 0.01), but in our WGS analysis, they associate with $P < 1 \times 10^{-60}$ in UK Biobank and $P < 1 \times 10^{-17}$ in All of Us. This is, at least partly, explained by poor coverage of some of the exons in *LCORL* from exome sequencing. Other explanations include poor array-based genotyping/imputation of rare variants, poor quality exome sequencing in GC-rich exons, lack of methods for meaningful aggregate testing, etc.

There are a number of limitations to our study. First, we did not explore physical colocalization of common and rare genetic associations for WHRadjBMI for two primary reasons. In contrast to height and to a lesser degree BMI, we identified relatively few rare genetic associations for WHRadjBMI. Also, more common SNP-based heritability remains to be explained in all broad genetic ancestry groups. Consequently, rare genetic associations may reside close to common variant associations that are yet to be detected by larger GWAS efforts. Second, both UKB and AoU are population cohorts that are affected by healthy volunteer recruitment bias and may therefore be depleted of individuals with more extreme phenotypes. Third, for the all-ancestry UKB meta-analysis, we were unable to classify independent genetic loci due to the high-risk of confounding associated with population stratification of genetic data involved in pooled cross-ancestry reference panels, particularly for rare variants. As we demonstrate in this article, and has been shown previously[37], conditioning non-coding aggregates on nearby causal coding associations is essential for interpretation. Finally, WGS data were generated using short-read sequencing, so information on structural variants and haplotypes was missing. Future studies of more diverse ancestries and using long-read sequencing technologies are likely to add further insights to the field. We additionally did not consider individuals of non-EUR UKB participants due to sample size constraints.

In summary, we have shown that WGS enables the discovery of associations with rare variants not found by other technologies. We have found coding associations for several anthropometric traits in genes with poor exome-sequencing coverage, and non-coding associations near genes for which no coding association could be identified. Furthermore, we have shown that common, rare and aggregate associations for height converge on the same loci, suggesting shared underlying biology.

## Methods

### UK Biobank and all of us whole genome sequencing

The whole genome sequencing performed for UKB had an average coverage of 32.5X using Illumina NovaSeq 6000 sequencing machines[38]. The genome build used for sequencing was GRCh38: single variant nucleotide polymorphisms and short 'indels' were jointly called using DRAGEN 3.7.8.

We set any UKB-WGS genotype calls to missing if either the sum(LAD) < 8 (local allele depth; LAD) per sample-genotype or GQ < 10 (genotype quality; GQ) for each of the 154,430 pVCFs provided by UKB using bcftools[39]. After these additional quality control steps, the transmission rate of singletons in 100 randomly selected pVCFs

covering 2,000,000 base pairs, which should theoretically be exactly 0.5 (assuming the majority of variants are not under strong negative selection), was 0.497, as compared to 0.456 as originally provided by UKB.

The whole-genome sequencing performed for AoU had average coverage ≥30X using Illumina NovaSeq 6000 sequencing machines. The genome build used for sequencing was GRCh38: single variant nucleotide polymorphisms and short 'indels' were jointly called using DRAGEN 3.4.12.

We exported the All of US WGS data from the provided VDS format to VCF format using HAIL v0.2.126. We subsequently set genotype calls to missing if 'FILTER!=PASS' (AoU site level filtering; FILTER), and, as with the UKB WGS data, set REF/ALT and ALT/ALT genotype calls to missing if either the sum(LAD) < 8 or GQ < 10 per sample-genotype, using bcftools[39]. We were unable to filter homozygote reference calls in the AOU WGS data equivalently because their LAD values were not provided, and their GQ values were not provided with the resolution required.

For both All of Us and UK Biobank, we subsequently dropped any variant with a missingness greater than 10%. Indels were normalised and left-aligned using bcftools based on a 1000 Genomes b38 reference available at https://ftp.1000genomes.ebi.ac.uk/vol1/ftp/technical/reference/GRCh38_reference_genome/ (accessed 30/03/2024). Finally, a multi-allele splitting procedure was applied, and each variant was assigned a unique ID (CHR:BP:REF:ALT) before merging all VCFs per chromosome. Each merged pVCF was then converted to plink[40] (v2.0) p(gen/var/sam) format.

### Genetic variant annotation
We annotated all genetic variants using Ensembl Variant Effect Predictor (VEP)[18] v110, LOFTEE[41] and UTRannotator[42]. Where possible, we assigned each variant to one of three *classifications*: coding, proximal-regulatory or intergenic-regulatory. A variant was classified as coding if it had a predicted impact on the coding sequence of **any** transcript; proximal-regulatory if the variant lay within a 5 kbp window of the UTRs of a transcript, and was not already a coding variant in any transcript, and finally intergenic-regulatory if it was not coding, but we did not assume proximity to coding transcript (see below). We additionally tested variants in sliding windows of size 2000 base pairs, regardless of the number of variants in each window, with coding and proximal-regulatory variants excluded to minimise hypothesis-testing overlap.

We then assigned each variant to groupings, which we refer to as *masks*, according to their predicted consequence and location. We used five published variant scores to group variants by consequence:

### Genomic evolutionary rate profiling (GERP)
The GERP score is a measure of conservation at the variant level[20]. We classified a variant as highly conserved if it had a GERP score >2.

### phastCon score
phastCon is a window-based measure of conservation across species[43]: either strictly mammalian (phastCon 30), or for all species (phast_100). We tested non-coding genome windows, i.e., excluding any window containing an exon, that had a phastCon score in the 99th percentile.

### Constraint score (gnomad 1 kb windows)
Constraint was calculated in windows of size 1 kbp[33] based on the local mutability and observed mutation rate of each window.

### SpliceAI score
The SpliceAI score[44] is a measure of how well predicted each variant within a pre-mRNA region is of being a splice donor/acceptor, or neither. A variant was classified as a splice site with high confidence if it had an AI > 50.

### Combined Annotation Dependent Deletion score (CADD)
The CADD score[21] predicts how deleterious a variant is likely to be. We applied the CADD score only to coding variants and considered loss-of-function variants only if tagged as high confidence by VEP. Missense variants with CADD > 25 were segregated for testing in a separate mask.

### JARVIS Score
The JARVIS score[19] was derived to better prioritise non-coding genetic variation for association study, based on a machine learning model derived from measures of constraint.

Each genome mask consisted of a number of variants with different *consequences*, based on their location, one of the above scores and/or predicted coding consequences. For example, for a variant to be classified as missense CADD > 25, it must change a codon of an exon of a gene transcript and be predicted to be highly deleterious.

In **ST24**, we present the full list of consequences assigned to each mask and classification.

### Association analyses
We performed both single variant and aggregate tests genome-wide for each of the three anthropometric phenotypes. All discovery association analyses were corrected for age, sex, age squared, UK Biobank recruitment centre, the first forty genetic principal components and whole-genome sequencing batch. Replication association analyses were corrected for age, sex, age squared, whole-genome sequencing batch and the first sixteen genetic principal components.

### Single variant association testing
To identify single variants associated with each phenotype we first performed an association test for all genetic variants with a minor-allele count of at least 5 using *REGENIE*[17] (v3.3) within 1361 pseudo-linkage-independent chunks[45] of the 22 autosomes, the two pseudo-autosomal (PAR; PAR1 & PAR2) regions of chromosome X and 100 equally sized chunks of the non-PAR regions of chromosome X. Chunk-wise lead variants were then selected in a conditional-joint analysis using an altered version of *GCTA-COJO*[22] (with command line options diff-freq = 0.2, COJO-P = $2.95 \times 10^{-10}$), with the UK Biobank whole-genome sequencing data, limited to individuals with each phenotype, as an LD reference panel.

Testing revealed that *GCTA-COJO* filters variants if their variance explained is >900-times the smallest variance explained by any independent variant ("sqrt(ldlt_B.vectorD().maxCoeff() / ldlt_B.vectorD().minCoeff()) > 30": line 732 of gcta/meta/joint_meta.cpp at https://github.com/jianyangqt/gcta (accessed 17/03/2024)). We understand that this filter exists to capture statistical confounding caused by collinearity, for example, if the reference genome used to calculate LD and the genetic data do not correlate well. However, for our purposes of jointly considering common and rare variants, where we have used the exact LD-reference panel matching our phenotype-specific discovery data set, we found that this filter was falsely removing large-effect variants. We thus removed this filter and recompiled *GCTA-COJO*, which is available at https://github.com/ExeterGenetics/WGS_50k_Proteins_2024.

To define chromosome-wide independently associated variants, we applied a second round of the altered *GCTA-COJO* algorithm, considering only those variants that were classified as independent at the chunk-level.

### Rare variant genomic aggregate testing
To identify coding and regulatory regions of the genome that were insufficiently powered for single variant analysis, we subsequently performed rare-variant (minor-allele frequency <0.1%) genomic aggregate association tests using the annotations described in Supplementary Data 24.

To test whether rare variant aggregate signals were caused by/confounded by residual LD and haplotype structure with common variants and or single variant signals, we performed the following steps for each rare variant aggregate test result reaching *Bonferroni P < 0.05*:

1. **To generate our primary discovery results** we adjusted for the common lead variants identified as independent signals in the joint (COJO) analysis (at MAF > 0.1%).
2. **To identify independent aggregate associations** if at least one aggregate passed our significance threshold we performed a forward stepwise regression. Starting from the most-strongly associated aggregate (by p-value), we performed an additional aggregate-testing run on aggregates reaching genome-wide significance, adjusting for all variants in the top signal. This process is repeated, with more variants added from the next most strongly associated aggregate, until no aggregate is genome-wide significant.

*TTN* was excluded from our conditional step-wise approach for aggregates due to regression convergence issues, due to its size.

### *REGENIE* performs four types of genome unit tests:

1. Standard BURDEN tests, under the assumption that each variant in a given gene unit mask has approximately the same effect size and sign on the phenotype.
2. SKAT tests, where the sign of the association of each variant in the unit is allowed to vary.
3. ACAT tests, where the sign of association of each variant in the unit can differ, and only a small number of variants in the mask need to be associated.
4. ACAT-O, which is an omnibus test of BURDEN, SKAT and ACAT that aims to maximise the statistical power across the three tests.

We performed each of the four statistical tests above for each mask for which a gene unit has at least one variant. Additionally, an association test was performed for all singleton variants (with MAC = 1) in each unit. *REGENIE* also estimated an 'all-mask' association strength for each genome unit, which is an aggregation of the test statistics of the individual masks. To ensure that this did not result in a mixing of non-coding and coding association statistics, we split each gene transcript into a coding transcript, which we tested for all coding masks, and a proximal transcript that we tested for all proximal masks. Regulatory genome units were either classified by their ENSR assignment, by the extent of a 1kb gnomad constrained window, or by a phastCon conserved window. We named sliding window masks by the region of the respective chromosome that they covered.

### UKB meta-analysis
To meta-analyse single variant and burden-type aggregates, we used GWAMA[46] under a fixed-effects model. For variance-based aggregates tests (SKAT, ACAT, ACAT-O), we performed a weighted (by sample size) *p*-value meta-analysis in R v4.1.2 using the ACAT[47] package.

### Statistical significance
Statistical significance was defined based on the minimum p-value observed for a whole-genome sequencing analysis of 20 randomly generated normally distributed continuous traits. The minimum p-value for single variant and aggregate association analyses was treated as independent: $P$ (single variants) $= 2.95 \times 10^{-10}$; $P$ (aggregates) $= 8.71 \times 10^{-9}$.

### Adjustment for previously reported GIANT loci
For each independent single variant and aggregate association, we performed a second round of adjustment for previously reported GIANT GWAS common-variant and exome-chip loci for each of the three anthropometric traits[2,23–27]. The effect sizes reported in the main results section are those calculated after adjustment for these previously reported loci—not those reported from the GCTA-COJO analyses. Locus zoom plots showing association statistics before and after adjustment for GIANT loci employed locuszoomr[48].

### Heritability
We calculated heritability using RHE-mc[49] applied to UKB WGS data for $N = 343,451$ maximally-unrelated European individuals, using variants with a minor-allele frequency greater than or equal to 1e-4. Maximally unrelated individuals were determined as the set of UKB-EUR-WGS for whom a GRM using all variants with MAF > 0.1% had an off-diagonal component <0.05 (related individuals were removed in a way to maximise total sample size). For our primary analysis, we stratified the heritability into three minor-allele frequency bins: $1 \times 10^{-4} < MAF \leq 1 \times 10^{-3}$, $1 \times 10^{-3} < MAF \leq 1 \times 10^{-2}$ and $MAF > 1 \times 10^{-2}$). We additionally split variants into three bins of LD-score, calculated using GCTA[22] (--ld-score --ld-wind 1000 --ld-rsq-cutoff 0.01), LD-score≤ 25th percentile, 25th percentile <LD-score ≤ 75th percentile; and LD-score > 75th percentile. Where appropriate, we additionally split variants based on their inclusion within a 35 kbp window surrounding known GWAS variants. In all cases, before running RHE-mc, we rank-inverse-normalised each trait separately for each sex, and residualised on the following covariates: sex, age, age squared, centre, sequencing centre and genetic principal components 1–40. To minimise the risk of population stratification inflating our heritability estimates, we additionally adjusted for 100 haplotype components, calculated using SparsePainter[50,51].

To replicate the results of Wainschtein et al. (2022), we used a random subset of $N = 24,932$ individuals from the maximally unrelated set of Europeans. We split variants into 4 MAF bins and 3 LD tertiles following the methodology of Wainschtein et al. (2022). We calculated the GRMs and GREML heritability estimates with GCTA (gcta64 --make-grm-bin --make-grm-alg 1; gcta64 --reml --reml-no-constrain), using the same phenotype and covariates as our primary analysis. As a comparison, we calculated heritability using RHE-mc in the subset sample with identical variant bins. We also used RHE-mc to calculate heritability in the primary sample using the 4 Wainschtein et al. (2022) MAF bins, with 3 LD tertile bins and also with 3 LD bins as defined in our primary analysis.

As a secondary analysis, applied exclusively to height, we calculated heritability based on 17 non-overlapping minor-allele frequency bins, from a minimum minor-allele count of 10, with the following upper-bounds (inclusive): $1.7 \times 10^{-5}$, $2.2 \times 10^{-5}$, $3.7 \times 10^{-5}$, $5.9 \times 10^{-5}$, $1.0 \times 10^{-4}$, $2.0 \times 10^{-4}$, $4.0 \times 10^{-4}$, $1.0 \times 10^{-3}$, $5.0 \times 10^{-3}$, 0.01, 0.02, 0.04, 0.1, 0.2, 0.3, 0.4, and 0.5.

### Ethics
This research complies with all appropriate ethical regulations. Ethics approval for the UKB study was obtained from the North West Centre for Research Ethics Committee (protocol no. 11/NW/0382). Informed consent for all All of Us participants is conducted in person or through an eConsent platform that includes primary consent, HIPAA Authorisation for Research use of EHRs and other external health data, and Consent for Return of Genomic Results. The protocol was reviewed by the Institutional Review Board (IRB) of the All of Us Research Programme. The All of Us IRB follows the regulations and guidance of the NIH Office for Human Research Protections for all studies, ensuring that the rights and welfare of research participants are overseen and protected uniformly.

### Reporting summary
Further information on research design is available in the Nature Portfolio Reporting Summary linked to this article.

## Data availability

Individual-level data cannot be shared publicly because of data availability and data return policies of the UK Biobank and All of Us. Data are available from the UK Biobank and All of Us for researchers who meet the criteria for access (http://www.ukbiobank.ac.uk; https://allofus.nih.gov/). Genome-wide summary statistics for both the UKB-EUR and UKB-meta-analyses are available at https://zenodo.org/records/17775928. Source data, where sharable, for figures is available at https://doi.org/10.5281/zenodo.18153043 (ref. 52).

## Code availability

The computational framework used for our WGS association analyses is available at https://doi.org/10.5281/zenodo.14204727 (ref. 53), and the plotting code is available at https://doi.org/10.5281/zenodo.18153043 (ref. 53).

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

## Acknowledgements

GH is supported by the Medical Research Council grant UKRI327. HW and MNW are supported by Medical Research Council grant MR/Y003748/1. ARW is supported by the Academy of Medical Sciences / the Wellcome Trust / the Government Department of Business, Energy and Industrial Strategy / the British Heart Foundation / Diabetes UK Springboard Award [SBF006\1134]. TMF is supported by MRC awards MR/WO14548/1 and MR/T002239/1. The research utilised data from the UK Biobank resource, carried out under UK Biobank application number 103356. UK Biobank protocols were approved by the National Research Ethics Service Committee. The authors would like to acknowledge the use of the University of Exeter High-Performance Computing (HPC) facility in carrying out this work, funded by an MRC Clinical Research Infrastructure award (MRC Grant: MR/M008924/1). This study was supported by the National Institute for Health and Care Research Exeter Biomedical Research Centre. The views expressed are those of the authors and not necessarily those of the NIHR or the Department of Health and Social Care. We gratefully acknowledge All of Us participants for their contributions, without whom this research would not have been possible. We also thank the National Institutes of Health's All of Us Research Programme for making available the participant data examined in this study.

## Author contributions

M.N.W., A.R.W., C.F.W., K.C., and G.H. jointly conceived the study. G.H., H.I.W.W., R.N.B., K.C., A.R.W., and M.N.W. conducted the analyses and wrote the manuscript. A.I.H., L.J., A.M., K.P., and T.M.F. edited the manuscript. All authors have read and approved the manuscript.

## Competing interests

The authors declare no competing interests.
