## [Transparent Peer Review file · Nature Communications]

Whole-genome sequencing analysis of anthropometric traits in 672,976 individuals reveals convergence between rare and common genetic associations

Corresponding Author: Dr Gareth Hawkes

Version 0:

Reviewer comments:

Reviewer #1

(Remarks to the Author)

Review of Hawkes et al, "Whole-genome sequencing analysis of anthropometric traits in 672,976 individuals reveals convergence between rare and common genetic associations".

Hawkes and colleagues have submitted an interesting whole-genome sequencing study of anthropometric traits in 672,976 individuals from the UK Biobank and All of Us. The stated purpose of the study was to assess the contribution and architecture of rare non-coding variants to three commonly studied anthropometric traits: height, BMI, and WHRadjBMI. They conducted the study using a two-stage design of discovery in 447,461 individuals in UK Biobank and replication in 225,515 individuals in All of Us. Across all traits, they identified 90 novel rare and low-frequency single variant associations. The Authors also showed that for height only, 97% of rare variant associations occur near GWAS loci demonstrating convergence of rare and common variant associations. They also reported that very rare variants explain less heritability than common variants and is similar across ancestral groups.

While the reporting findings are exciting and important, I have several major concerns as follows:

1. I strongly urge the Authors to redo the analyses including all persons with data in UKBiobank. Recently published papers urge scientists to focus attention on pooled- or meta-analysis results of all participants (Recommendations on the use and reporting of race, ethnicity, and ancestry in genetic research: experiences from the NHLBI Trans-Omics for Precision Medicine (TOPMed) program). The authors justify the exclusion of populations due to sample size but I completely disagree. At the very minimum they could have done an all-persons analysis followed by a European population only analysis. Although the Authors should consider potential benefits versus potential harms when conducting population-specific analysis. Also there is a large disconnect between their approach to UK Biobank and All of Us.
2. In the discussion, the Authors highlighted interesting findings for IGF2BP2, UBR3, and FGF18. These deep dives are very interesting. However, there is very limited discussion of the biology of the other novel loci. This is a missed opportunity.
3. I also think that the physical colocalization of common and rare genetic associations for BMI should have been considered. While I somewhat agree with their reasoning for WHRadjBMI I do not agree for BMI and would like to see the results from these analyses.

Reviewer #2

(Remarks to the Author)

In this manuscript (Hawkes et al.), the authors performed single-variant and aggregate association tests in European

population in UK Biobank WGS for anthropometric traits with focusing on rare-variant associations. They sought to replicate their findings in multi-ancestral populations in All of Us WGS. They reported novel independent associations with potential functional interpretations, and assessed heritability explained by rare variations. They have already published rare-variant association study using ~200K individuals, and this is an updated study using ~450K individuals.

I think the analyses were thorough and it was nice to see the quantitative assessment of novel rare-variant associations independent from common variants' associations. There were several points that I thought were a little unclear in the presentation or interpretation of the results. I will summarize them as specific major comments below.

1. Total heritability and heritability explained by rare variants: I think it would be valuable to compare the authors' results with Weinschtein et al. 2022 (PMID: 35256806) using different dataset (with a smaller sample size) and a statistical method (GREML). In particular, it was difficult to interpret the results regarding heritability explained by 17 MAF bins shown in Fig 4c. How these estimates change if the authors were to take similar statistical approaches to Weinschtein et al. with more variants in the bins as defined by them? I also wanted to see the reasons for the inflated total heritability values in these analyses that the authors had pointed out.
2. Replication in AoU: Did the percentage of replication with $P < 0.05$ account for directional concordance of effect estimates? If not (as might have been suggested in Fig 1 AFR), I think it should be accounted for and the percentage should be recalculated.
3. Novel non-coding associations: I recognize that it is non-trivial to annotate non-coding variations with their target genes or functional units. In the examples such as MIRNA497, IGF2BP2 and FGF18 where the authors annotated the rare associations with gene names, the strategies for annotations (e.g., take the closest genes / no other genes within XXkb nearby etc.) will still be useful for the readers. Adding locus-zoom plot with gene annotations for MIRNA497 might be useful. I also thought that the raw genome browser view in Fig 2A was a little hard to read.
4. Aggregate associations: While I acknowledge that the authors summarized the methods for aggregate association tests in the Method section, it would be helpful to briefly explain them in the main text as well when the results were introduced. Readers might be curious to know the brief summary of aggregation strategy based on variants' annotations and also statistical methods (burden/SKAT/...) used for aggregation for the loci with significant associations.
5. Data Availability: The section mentioned that the data cannot be shared publicly, but I understand that summary statistics for single-variant and aggregate association tests can and should be shared publicly to the community.

Minor Comments:

1. There were two method sections – 'ONLINE METHODS' and 'Methods', which I am not sure if they may want to reorganize when finalizing the manuscript.
2. Introduction. "~40% of the variance among individuals of European genetic ancestry": If I am correct, this number seems slightly different from what I found in the heritability estimate in Yengo et al. 2022.
3. Introduction. "Rare non-coding variants may explain a substantial amount of the remaining heritability, but their contribution is largely unexplored": As I raised in the major comments, these are potentially over-statement, since there are several previous studies (like Weinschtein et al. and the one by the authors themselves) with similar aims using WGS. It should be valuable to cite and discuss them with the authors' results as comparison.
4. Introduction. "Identifying rare non-coding associations may also substantially aid the identification of the causal gene at GWAS loci": I think it is generally challenging to link rare non-coding variations to their target genes. What's the authors' rationale to claim that rare non-coding associations could 'substantially aid'?

Version 1:

Reviewer comments:

Reviewer #1

(Remarks to the Author)

1. In Point 1, my original statement was: "Recently published papers urge scientists to focus attention on pooled or meta-analysis results of all participants (Recommendations on the use and reporting of race, ethnicity, and ancestry in genetic research: experiences from the NHLBI Trans-Omics for Precision Medicine [TOPMed] program) ... At the very minimum, they could have conducted an all-persons analysis followed by a European population-only analysis." To be clear, I was not advocating for a pooled analysis. When I referred to an "all-persons analysis," I meant a meta-analysis. For rare variants, pooled analyses are methodologically inappropriate—something the authors themselves illustrate in their work.

The Editors must determine whether excluding populations from a study based on ancestry is justifiable. While the sample size is modest, 10,000 participants is more than adequate to merit serious consideration. At the very least, this could be included as a secondary analysis in the supplementary materials, rather than serving as the primary analysis on which conclusions are based.

2. I stand by my earlier observation that while the discussion provides valuable deep dives into IGF2BP2, UBR3, and FGF18, there is minimal discussion of the biology underlying the other novel loci. This omission is a missed opportunity to offer richer scientific context and strengthen the manuscript's impact.

3. In addition, the potential physical colocalization of common and rare genetic associations for BMI should have been examined. While the authors' reasoning for excluding WHRadjBMI analyses is somewhat acceptable, their justification for BMI is not. A much larger polygenic risk score (PRS) for BMI has now been published (PMID: 40691366), rendering their reasoning outdated. These analyses should be repeated with the updated PRS, and the results presented, to ensure the conclusions remain valid.

Reviewer #2

(Remarks to the Author)

I thank the authors for responding to my previous comments and conducting additional analyses. I appreciate that all my comments were adequately addressed and the manuscript has improved.

Version 2:

Reviewer comments:

Reviewer #1

(Remarks to the Author)

Should be good to go.

We would like to thank the reviewers for taking the time to review our manuscript, and for providing considered and insightful comments on our work. We have responded to their comments below, and we hope they find the changes satisfactory.

REVIEWER COMMENTS

Reviewer #1 (Remarks to the Author):

Review of Hawkes et al, “Whole-genome sequencing analysis of anthropometric traits in 672,976 individuals reveals convergence between rare and common genetic associations”.

Hawkes and colleagues have submitted an interesting whole-genome sequencing study of anthropometric traits in 672,976 individuals from the UK Biobank and All of Us. The stated purpose of the study was to assess the contribution and architecture of rare non-coding variants to three commonly studied anthropometric traits: height, BMI, and WHRadjBMI. They conducted the study using a two-stage design of discovery in 447,461 individuals in UK Biobank and replication in 225,515 individuals in All of Us. Across all traits, they identified 90 novel rare and low-frequency single variant associations. The Authors also showed that for height only, 97% of rare variant associations occur near GWAS loci demonstrating convergence of rare and common variant associations. They also reported that very rare variants explain less heritability than common variants and is similar across ancestral groups.

While the reporting findings are exciting and important, I have several major concerns as follows:

1. I strongly urge the Authors to redo the analyses including all persons with data in UKBiobank. Recently published papers urge scientists to focus attention on pooled- or meta-analysis results of all participants (Recommendations on the use and reporting of race, ethnicity, and ancestry in genetic research: experiences from the NHLBI Trans-Omics for Precision Medicine (TOPMed) program). The authors justify the exclusion of populations due to sample size but I completely disagree. At the very minimum they could have done an all-persons analysis followed by a European population only analysis. Although the Authors should consider potential benefits versus potential harms when conducting population-specific analysis. Also there is a large disconnect between their approach to UK Biobank and All of Us.

Thank you for your comments: we agree that this is an incredibly important issue. However, we would like to clarify that our concern about sample size is secondary to, and caused by, our reluctance to consider rare variants in the full sample size containing individuals with significantly differing genetically-inferred ancestry. As can

be seen in heritability analyses, which were restricted to unrelated European individuals only, we still see substantial evidence of (amongst other possible factors) population stratification even for variants at minor-allele-count ≥ 10 . Only once we had made the choice to stratify the rare-variant analyses by genetically-inferred ancestry, the stratified sample sizes are too low. The issue of population stratification for rare variants, even after adjusting for PCs, is discussed more here: PMID32183706.

Our strategy of performing meta-analyse across genetic ancestries in All of Us was specifically to ensure that our discovery findings were generalisable and enabled by the fact that All of Us contains substantially more individuals of inferred non-European genetic ancestry for rare variant association testing (N_AOU AFR = 54,940 vs. N_UKB AFR <10k ; N_AOU AMR = 42,009 vs N_UKB SAS <10k).

Nevertheless, we have now performed a combined single-variant analysis of all UKB samples as requested (UKB-ALL). We find that the number of 'independent' rare (MAF<1%) genetic variants increases substantially for WHRadjBMI: 32 independent variants for all samples vs. 10 for EUR-only (220% increase), and BMI: 24 vs. 14 for EUR-only (71% increase). For these two traits, the nominal replication rate ($P < 0.05$ and matching effect-size sign) for rare variants reduced to ~39% (14/36), and for Bonferroni-corrected replication to ~11% (4/36). This is in comparison to 75% and 38% in our original analysis, respectively.

Subsequently, we have assessed the degree to which the all-sample inclusion increased the effects of population stratification for all SNPs in our analysis, not just CoJo-selected variants. We used height as an exemplar trait given known differences in average height across populations, including along the north and south axis of Europe where, on average, northern Europeans are taller than southern Europeans. First, using data from the 1000 Genomes Project (1000GP), we tested for a relationship between the average difference in allele frequencies of height raising alleles (based on our UKB-ALL analysis) between individuals of broad European (EUR) genetic ancestry and either South Asian (SAS) or African (AFR) genetic ancestry, and the strength of association. Our hypothesis was that in the absence of population stratification we should see no relationship between strength of association and differences in allele frequencies.

Based on our primary European-based UKB analysis, we did not observe a relationship between strength of associations and the difference in allele frequencies between northern and southern Europe, represented by individuals from the 1000GP labelled from Great Britain and Tuscany, respectively.

EUR UKB Association Analysis

We found that variant p-values were significantly associated with allele-frequency difference in the all-sample analysis for both SAS-EUR differences ($P = 3e-9$) and AFR-EUR differences ($P = 3e-8$). – see below Figures.

All ancestries analysis

Second, we estimated and compared the variance in effects with $P < 0.05$ in the UKB-EUR and UKB-ALL analysis that could be explained by PC loadings based on data from the 1000 Genomes Project (1000GP). Our hypothesis is that uncorrected population stratification would significantly induce correlations between PC loadings and effect estimates, with more variance likely to be explained among rarer variants. Therefore, we performed both global- and ancestry-specific principal component analyses using the 1000GP data based on variants with $P < 0.05$ from UKB-EUR and UKB-ALL analyses separately.

Compared to our primary UKB-EUR analysis, the graphs below show that more variation in effect estimates derived from the UKB-ALL analysis can be explained by

PC loadings of the respective variants, specifically based on the global and “AFR” principal component space, accentuated when only considering variants with $0.1\% < \text{MAF} < 1\%$ from the UKB-ALL analysis. Given the relative size of the 1000GP dataset, we were unable to examine variants rarer than 0.1% in UKB where we expect additional gains in variation explained due to uncorrected population stratification.

X-axis: the number of principal component loading vectors incrementally added to a multivariable linear regression model in order of the principal component ranking. Y-axis: the variance in the effect estimates explained by the loading vectors in the respective model based on (adjusted) r^2 from the model. Betas were aligned to alleles the PC loading corresponded to. Dashed lines represent results based on data from UKB-EUR (where only 1000GP-ALL and 1000GP-EUR PC spaces were generated for analysis). Solid lines represent results based on data from UKB-ALL. Different colours represent different broad ancestry strata of the 1000GP. Closed and open circles represent multivariable models with $P_{F\text{-statistic}} < 0.05$ and > 0.05 , respectively.

With these results in mind, we believe it is best to maintain our current analysis structure, with a strong emphasis that our methodology in All of Us aims to maximise the equity of our findings. We have, however, added text to the Online Methods expanding the thinking behind our choice “As our interest was in novel rare variants, we did not consider the two second-largest ancestries within UKB due to their small sample size (South Asian; UKB-SAS & African; UKB-AFR; $N < 10,000$). **We decided against a cross-ancestry pooled approach in UK Biobank due to concerns regarding population stratification, particularly for rare variants (PMID 32183706).** Instead, to demonstrate the generalisability of our findings beyond EUR-individuals, we replicated our results in the diverse AoU Cohort, based on individuals of inferred European (AoU-EUR; $N = 128,566$), African (AoU-AFR; $N = 54,940$) and admixed-American (AoU-AMR; $N = 42,009$) (bolded new).”

If the reviewer would prefer, we could include these results in our Zenodo data release as supplementary data.

2. In the discussion, the Authors highlighted interesting findings for IGF2BP2, UBR3, and FGF18. These deep dives are very interesting. However, there is very limited discussion of the biology of the other novel loci. This is a missed opportunity.

We understand the reviewers concern, but we made the choice to focus on those loci because they provide the clearest examples of the type of finding that can be made from WGS aggregate testing rather than other approaches. We also wanted the paper to focus more on the results from our heritability and co-localisation analyses.

3. I also think that the physical colocalization of common and rare genetic associations for BMI should have been considered. While I somewhat agree with their reasoning for WHRadjBMI I do not agree for BMI and would like to see the results from these analyses.

We thank the reviewer for this suggestion. As mentioned in the manuscript, we were reluctant to undertake this analysis for BMI and WHRadjBMI given the common SNP-based heritability has not been saturated in the same way as it has been for height. Therefore, estimates of distances between rare variants to published common SNPs are likely to be inflated given the absence of common associations yet to be identified through larger studies with power to report them. However, we have added additional supplementary tables (now ST15 and ST16) to show the physical colocalization of the 4 variants and 2 aggregate associations with BMI in the UKB-EUR analysis surviving adjustment for previously reported BMI variants and associated in AOU-EUR at $P < 0.05$ to the nearest common variant using results from the largest BMI GWAS meta-analysis published to date (Yengo *et al.* 2018). We have also added the following text to the results section:

For BMI, we observed similar levels of physical proximity for four autosomal variants reaching genome-wide significance after adjustment for previously published BMI associations by the GIANT consortium and reaching $P < 0.05$ in AOU-EUR. All four variants (including two with UKB-EUR MAF $< 1\%$) were within 35kb of 941 common variants previously associated with BMI²⁵ (Supplementary Table 15). However, two aggregate-based associations identified in UKB and with AOU-EUR $P < 0.05$ were located further away from the nearest common BMI GWAS SNP ($> 1.7\text{Mb}$) (Supplementary Table 16).

Reviewer #2 (Remarks to the Author):

In this manuscript (Hawkes et al.), the authors performed single-variant and aggregate association tests in European population in UK Biobank WGS for anthropometric traits with focusing on rare-variant associations. They sought to replicate their findings in multi-ancestral populations in All of Us WGS. They reported novel independent associations with potential functional interpretations, and assessed heritability explained by rare variations. They have already published rare-variant association study using ~200K individuals, and this is an updated study using ~450K individuals.

I think the analyses were thorough and it was nice to see the quantitative assessment of novel rare-variant associations independent from common variants' associations. There were several points that I thought were a little unclear in the presentation or interpretation of the results. I will summarize them as specific major comments below.

1. Total heritability and heritability explained by rare variants: I think it would be valuable to compare the authors' results with Weinschtein et al. 2022 (PMID: 35256806) using different dataset (with a smaller sample size) and a statistical method (GREML). In particular, it was difficult to interpret the results regarding heritability explained by 17 MAF bins shown in Fig 4c. How these estimates change if the authors were to take similar statistical approaches to Weinschtein et al. with more variants in the bins as defined by them? I also wanted to see the reasons for the inflated total heritability values in these analyses that the authors had pointed out.

As suggested, we have performed secondary analyses using both RHEmc and GREML (see Figure below), using the LD-tertile bin definitions from Wainschtein et al. We estimated that GREML would require >2TB of memory for the full EUR sample, which was not computationally feasible on the DNANexus RAP, which was one of our motivations for using the RHEmc (GENIE) methodology. Therefore, to make a direct comparison between GREML and RHEmc, we have down-sampled to 25,000 individuals. We additionally ran RHEmc on the full sample using LD-tertiles ('RHEmc 343k tertiles') and with three LD bins defined, similarly to our primary analysis, as the lower, two middle, and upper LD-quartiles ('RHEmc 343k quartiles').

Our down-sampled results using both GREML ('GREML 25k tertiles') and RHEmc ('RHEmc 25k tertiles') were concordant in every bin. The total heritability estimate was concordant between Wainschtein et al. 2022 and all our new analyses, as well as our primary analysis in Fig 4b. This provides further evidence for our assertion that confounding only becomes a major problem at $MAF < 1e-4$, as shown in Fig 4c and supplementary Fig 3.

We have added these comparisons into our results section, including a supplementary table, and added a methods section on GREML.

2. Replication in AoU: Did the percentage of replication with $P < 0.05$ account for directional concordance of effect estimates? If not (as might have been suggested in Fig 1 AFR), I think it should be accounted for and the percentage should be recalculated.

Thank you for pointing this out. We have corrected our figures to account for sign-consistency. We have also simplified our Bonferonni thresholds for consistency in the same section.

“Of the 119 novel rare and low-frequency variant associations which we could put forward for replication, 90 (75.6%) showed nominal evidence of replication ($P < 0.05$), and 42 (35.3%) showed strong evidence of replication at $P < 0.05/172$ with consistent effect-sizes. Of the 53 common variants with UKB-MAF $> 1\%$ put forward for replication, we observed 47 (88.7%) associations with nominal evidence of replication ($P < 0.05$), and 31 (58.5%) with Bonferroni-corrected evidence of replication ($P < 0.05/172$).”

3. Novel non-coding associations: I recognize that it is non-trivial to annotate non-coding variations with their target genes or functional units. In the examples such as MIRNA497, IGF2BP2 and FGF18 where the authors annotated the rare associations with gene names, the strategies for annotations (e.g., take the closest genes / no other genes within XXkb nearby etc.) will still be useful for the readers. Adding locus-zoom plot with gene

annotations for MIRNA497 might be useful. I also thought that the raw genome browser view in Fig 2A was a little hard to read.

We have improved the readability of Fig 2A: we hope this is now easier to interpret. In the case of the example non-coding associations, we have added the detail that *IGF2BP2* was the closest gene. For MIRNA497 and FGF18, we identified variants in the miRNA seed region and 5'UTR respectively, providing strong links.

4. Aggregate associations: While I acknowledge that the authors summarized the methods for aggregate association tests in the Method section, it would be helpful to briefly explain them in the main text as well when the results were introduced. Readers might be curious to know the brief summary of aggregation strategy based on variants' annotations and also statistical methods (burden/SKAT/...) used for aggregation for the loci with significant associations.

We have added some detail to the 'Online Methods': "*We applied three rare-variant aggregate testing procedures built into REGENIE: BURDEN, where all variants are assumed to act homogeneously on the phenotype; SKAT, where variants may act bi-directionally; and ACAT, where variants can act both bi-directionally, and some variants can have null estimates (see **Methods** for more details).*"

5. Data Availability: The section mentioned that the data cannot be shared publicly, but I understand that summary statistics for single-variant and aggregate association tests can and should be shared publicly to the community.

We will be making all summary statistics publicly available via Zenodo upon acceptance and publication – we have added a statement to the Data Availability section.

Minor Comments:

1. There were two method sections – 'ONLINE METHODS' and 'Methods', which I am not sure if they may want to reorganize when finalizing the manuscript.

As pointed out in your comment above, our methods are quite complex, so we felt it was beneficial to describe them here in addition to the primary methods. We can remove this at the editor's digression.

2. Introduction. “~40% of the variance among individuals of European genetic ancestry” : If I am correct, this number seems slightly different from what I found in the heritability estimate in Yengo et al. 2022.

Based on our understanding, this matches Yengo et. al 2022: based on the following from their abstract: “*In out-of-sample estimation and prediction, the 12,111 SNPs (or all SNPs in the HapMap 3 panel) account for 40% (45%) of phenotypic variance in populations of European ancestry but only around 10–20% (14–24%) in populations of other ancestries.*”

3. Introduction. “Rare non-coding variants may explain a substantial amount of the remaining heritability, but their contribution is largely unexplored” : As I raised in the major comments, these are potentially over-statement, since there are several previous studies (like Weinschtein et al. and the one by the authors themselves) with similar aims using WGS. It should be valuable to cite and discuss them with the authors’ results as comparison.

We have added a sentence to the introduction referencing existing WGS heritability studies (including Wainschtein et al). We have also added the direct comparison with Wainschtein et al into the results section, as described above.

4. Introduction. “Identifying rare non-coding associations may also substantially aid the identification of the causal gene at GWAS loci”: I think it is generally challenging to link rare non-coding variations to their target genes. What’s the authors’ rationale to claim that rare non-coding associations could ‘substantially aid’?

The main rationale for this is that rare variants usually have much bigger effects on the phenotype and usually have no or substantially fewer correlated variants compared to common variants at GWAS loci. So fine-mapping and subsequent functional studies become more tractable. The aggregate of multiple rare variants in prespecified annotation units also makes interpretation easier. In the simplest cases, such as *FGF18*, multiple independent non-coding 5’UTR variants link the GWAS loci with high-certainty to the causal gene. For *HMGA1*, which we highlighted in our previous paper (PMID 39362880), we identified an allelic series of rare variants in a *cis* enhancer each affecting height by ~5cm. There was also a common variant GWAS signal in the same enhancer (affecting height by <0.05cm). So we believe that, in combination, rare non-coding variants can substantially aid the identification of the variant, causal regulatory element and causal gene.

We would like to thank the reviewers for their comments which we have addressed below.

Reviewer #1 (Remarks to the Author):

1. In Point 1, my original statement was: “Recently published papers urge scientists to focus attention on pooled or meta-analysis results of all participants (Recommendations on the use and reporting of race, ethnicity, and ancestry in genetic research: experiences from the NHLBI Trans-Omics for Precision Medicine [TOPMed] program) ... At the very minimum, they could have conducted an all-persons analysis followed by a European population-only analysis.”

To be clear, I was not advocating for a pooled analysis. When I referred to an “all-persons analysis,” I meant a meta-analysis. For rare variants, pooled analyses are methodologically inappropriate—something the authors themselves illustrate in their work.

The Editors must determine whether excluding populations from a study based on ancestry is justifiable. While the sample size is modest, 10,000 participants is more than adequate to merit serious consideration. At the very least, this could be included as a secondary analysis in the supplementary materials, rather than serving as the primary analysis on which conclusions are based.

We apologise for the misunderstanding. As requested, we have now completed a meta-analysis of summary statistics from the three largest UKB genetic ancestries (European, African-Caribbean and South-Asian). Statistically significant aggregate associations from that analysis can be found in Supplementary Tables 12 & 13, which we reference in the results sections. We have also added to the online methods section of the manuscript:

“However, a meta-analysis of genetically inferred UKB-EUR (our primary discovery cohort), UKB-SAS and UKB-AFR was performed as a secondary analysis for comparison.”

Alongside the following expanded detail:

*“To identify conditionally independent genetic variants in our primary discovery analysis using UKB-EUR, we used a modified version of GCTA-COJO²² with the UKB-EUR sequencing data as the reference panel. To identify statistically independent rare-variant aggregates, we performed a forward-stepwise conditional approach (**Methods**). Using these methodologies, we were unable to identify conditionally independent associations for the secondary UKB-meta-analysis due to concerns regarding cross-ancestry pooling for LD-panels (Yengo et al 2022), particularly for UKB sequencing data applied to rare variant association discovery.”*

We additionally make light comparisons between the UKB-EUR and UKB-meta analysis in the results section. However, as described above and for the same reasons as described in our prior response regarding a pooled analysis, it is not possible for us to perform conditional or joint analysis on the rare-variant association results derived from this analysis. To our knowledge the methodology for such an analysis has not yet been developed. Given the importance of conditional analysis

for non-coding variation with respect to nearby coding associations that we highlight in the manuscript, this makes interpretation of association statistics difficult. We have added this caveat to the discussion section.

As stated in the manuscript, we will also include the full per-ancestry and meta-summary statistics via Zenodo.

2. I stand by my earlier observation that while the discussion provides valuable deep dives into IGF2BP2, UBR3, and FGF18, there is minimal discussion of the biology underlying the other novel loci. This omission is a missed opportunity to offer richer scientific context and strengthen the manuscript's impact.

These choices were made primarily due to the scope of the paper. The loci we have chosen to give detail on are designed to be exemplar, with the strongest evidence of replication or biological interpretation. Future publications may have the bandwidth to explore the other loci in more detail.

3. In addition, the potential physical colocalization of common and rare genetic associations for BMI should have been examined. While the authors' reasoning for excluding WHRadjBMI analyses is somewhat acceptable, their justification for BMI is not. A much larger polygenic risk score (PRS) for BMI has now been published (PMID: 40691366), rendering their reasoning outdated. These analyses should be repeated with the updated PRS, and the results presented, to ensure the conclusions remain valid.

Thank you for the suggestion – however we would like to note that we did complete the co-localisation for BMI in the previous revision, using the previously published Yengo loci. There was a leftover statement to the contrary in the discussion which has now been removed.

We were unable to update our analysis based on the recently published polygenic prediction paper for BMI because the list of conditionally independent variants derived through the described GCTA-COJO analysis used for one of the described predictors was not published. Only details of genome-wide polygenic predictors were made available by the authors. Furthermore, given this publication was released after our submission, we believe the current analysis is as extensive as possible.

Reviewer #2 (Remarks to the Author):

I thank the authors for responding to my previous comments and conducting additional analyses. I appreciate that all my comments were adequately addressed and the manuscript has improved.

Thank you for your time in improving our manuscript!